# Collaborative Design in Kinetic Performance: Safeguarding the Uilleann Pipes through Inertial Motion Capture

**Philip I. Morris \* and Joan Ramon Rodriguez-Amat**

Department of Media Arts and Communication, Sheffield Hallam University, Sheffield S1 1WB, UK
\* Correspondence: p.morris@shu.ac.uk

**Abstract:** This paper explores the possibilities of motion capture as a tool to preserve and study Intangible Cultural Heritage (ICH) practices such as playing instruments. The Uilleann pipes are both an instrument and a culture with a now strong community following and recovering after being almost extinguished several times in Ireland. The playing and making of the Uilleann pipes was incorporated into the Representative list of the UNESCO Convention of Intangible Cultural Heritage in 2017. This experience was performed in collaboration with several Uillean pipe players who contributed at every stage of the performance recording with comments and orientation. Some of these comments were also later formally kept as interviews to the players. The technical capture of the movements was done using a Rokoko Smart suit and paired Smart gloves which the piper wears for the performance. The resulting motion file was then cleaned and redirected to Blender 3D, a community made software package that allows the incorporation of a renderable avatar that helps for the dissemination of the performance. This recording process, called Inertial system, allows performers to perform almost anywhere and to capture the movements of the players with good accuracy. This process of recording and collaboration with the community is a cost-effective solution that can be of particular interest for scholars as well as for cultural communities wishing to archive these practices quickly. This design of technology and collaborative recording allows for a round experience that combined the detail of the technically enhanced recording with the interpretive nuance of the player that enriches the capturing process with mentions to how it is 'relatively' comfortable for the player and how efficient it is in terms of resulting detail. This collaboratively designed experience also provides the three aspects of intangible heritage preservation: supports the community -who can learn from the resulting animation; helps situating the practice within the cultural practice of the community-as they are part of the process; and helps materialise the output permitting its digital cataloguing, archival, and storage.

**Keywords:** collaborative design; motion capture; Intangible Heritage; Uilleann pipes; community

## 1. Introduction

Using technology to capture, study and preserve Intangible Cultural Heritage (ICH) has been done before. In fact, there are multiple techniques and approaches available in the literature that show a variety of examples. Starting with the use of recording devices that capture the lost sound of old symphonies, and continuing with videotapes recording choreographies, the line of technological efforts to capture the ephemeral is long, but it is still improving and evolving with the technological change: the possibilities of storage and scholarly research, and with the interaction with the community members.

This research project opens a new strand that consists of using motion capture and digital animation techniques to increase the precision of the recordings of intangible cultural practices. These motion recordings are captured within their cultural context and with the collaboration of their community members. This ensures the preservation and understanding of cultural practices that otherwise might go lost in the memory of the communities that practice them. Intangible Heritage, by its very nature, is dependent

on passing techniques which take many years to perfect from generation to generation. This dependence ultimately makes the practices quite tenuous and fragile. The role of a community in supporting Intangible Heritage and culture is vital and an asset for the local, national, and worldwide communities as well as it is a remarkable individual and cultural identity feature. Therefore UNESCO, rather belatedly, raised the instrument of the 2003 Convention for safeguarding Intangible Cultural Heritage.

The "Uilleann" or "union" pipes have been very culturally significant in Ireland since their evolution from the pastoral pipes around the mid-18th Century [1]. At about this time the Penal laws forbade the owning of musical instruments of many of the Irish citizens. However, the Uilleann pipes have proven to be very resilient thanks to communities within Ireland that have preserved the instrument through several historical existential crisis such as the famine in the mid-19th century. During that period, it was in no small part thanks to an emigrant population in places such as America and Canada who embraced the pipes, improved them, and popularised them [2]. The Uilleann pipes therefore are not only a product of a long Irish tradition, but their own technological transformation is witness of the hardships and wanderings that the Irish culture has historically endured. Studying the Uillean pipes and the community that practice and play them is a form of exploring the broader Irish culture and its practice.

In this case, the experience shared here shows how modern technology helps to capture the movements of an Uilleann piper and how this leads to the development of precise 3D digitally animated models that can then be stored and shared online. This is part of a broader project that included working together with the international community of pipers and incorporating the views of the participant during the process. This produced data shows the detailed movement of the player's body at playing but it also ensures a proper contextual immersion to his views on the cultural significance of his practice. The player's engagement in the process extended from the provision of orientation on the comfort of the equipment used for the motion capture, as well as the feedback on the sensations associated with their movements; to the analysis and review of the footage taken from the performance of another piper from a different area of Ireland. This engagement made of this initial study a collaborative experience, helped incorporate variations and improvements throughout the recording process, and has opened a significant strand for further research.

This article is a report of the collaborative experience of Motion Capturing the performance of a Uilleann pipe player and his views on the process and on the possibilities of this practice in the future and for the community shared memory. The literature review of this article is organised into three sections. First, a brief review of the Literature around the notion of Intangible Cultural Heritage and the United Nations safeguarding instrument with examples of where and how the use of technology and collaborative design has been used to capture elements of Intangible Cultural Heritage; second, a review of the main techniques of using Inertial Motion Capture with this setup, to situate the case. And third, the literature looks at the unique instrument of the Uilleann pipes itself [3]. The second part of this article deals with the specific experience of recording, processing, and analysing the playing of a piper with the incorporations of the contributions of the player himself.

## 2. Literature Review

Traditional ideas of heritage generally have been based around tangible artefacts such as the ones found in a museum or in a collection. To this end UNESCO ratified the Convention Concerning the Protection of the World Cultural and Natural Heritage in 1972. The Convention is an instrument that seeks to protect artefacts or sites with cultural value. Critics were quick to point out that there was a "*geographic imbalance among sites included in the world heritage list*" [4] specifically because it included mostly western sites and "*for almost four decades, UNESCO's normative standard-setting activities focused on the protection of tangible heritage*" [5]. This is a perspective criticised and challenged by scholars such as Laurajane Smith who argue that heritage, especially tangible, physical heritage "*is an act of communication and meaning making—indeed an experience*" [3].

The International community became aware, after the ratification of the Convention Concerning the Protection of the World Cultural and Natural Heritage in 1972 [6], that steps needed to be taken to Safeguard immaterial manifestations of culture which by their very nature are fragile, generational, and prone to disappear. Indeed, as early as 1964, already, the International Council on Monuments and Sites (ICOMOS) used the 2nd International Congress of Architects and Technicians of Historic Monuments, to mention that the architectural preservation of historic documents needed to be done "within the framework of its own culture and traditions" [7]. Thirty years later, the Nara Document on Authenticity (1994) included an article #7 that read: "*All cultures and societies are rooted in the particular forms and means of tangible and intangible expression which constitute their heritage, and these should be respected*" [8]. These two early documents (before and after the 1972 Convention) show some sensitive awareness of the importance of the Intangible Cultural Heritage (ICH) and the need to support its protection. It was finally in 2003 when UNESCO ratified the Instrument for the Convention for the Safeguarding of Intangible Cultural Heritage (ICH) [9]:

> "*Intangible heritage is made up of processes and practices and therefore needs another safeguarding approach and methodology than tangible heritage. It is fragile by its very nature and therefore much more vulnerable than other forms of heritage because it hinges on actors and social and environmental conditions*". [5]

This was an important step not only to protect elements of Intangible heritage but also to take steps to support communities who foster and preserve these intangible elements and practices as part of their identity performance as community.

Indeed, what is now named as "Intangible Heritage" refers to the "*immaterial manifestations of culture . . . as well as the most important vehicle of cultural diversity*" [10] and is responsible for the "*mainspring of cultural diversity and a guarantee of sustainable development*" [11]. These manifestations are expressed in many ways by many cultures; and it took some time and several legal instruments and documents designed to find a way to support Intangible Cultural Heritage (ICH) manifestations.

Since those early steps, ideas about what to include, preserve or study have grown to encompass more intangible expressions such as the human process of creating artefacts, playing instruments, dance, and sports. In fact, examples from many human endeavours and cultures are now being collected and examined, and steps are taken to safeguard or guarantee their fragile unbroken lineage.

Many authors agree that tangible expressions of heritage embody intangible components such as spiritual values, symbols, meanings, knowledge or know-how of craftsmanship and construction . . . tangible aspects are not necessarily of outstanding universal value [12,13]. Others go further and express that intangible and tangible expressions of culture are interdependent, like two sides of the same coin [5]. Munjeri argues that "*the protection of intangible heritage is a long struggle*" [12] and that the protection that tangible heritage has enjoyed for forty years was heavily weighted towards these physical manifestations in detriment of the living cultures within which the objects acquired meaning. Tangible expressions need a political, social context to manifest their value. Objects themselves have little value without the cultural context within which their meaning is built; and those meanings are reproduced through intangible practices. Those intangible mechanisms of reproduction of culture need physicality and tangible manifestations to be made incarnate and to be situated within a (living) community that grants their value. The insistence on objectual and material -tangible heritage protection- gives rise to 'static view of human cultures' [12] whereas in reality "*[human cultures are] complex and multidimensional, [...and each . . . ] individual piece of evidence should therefore be considered not in isolation but within its whole context with an understanding of the multiple reciprocal relationship that it had with its physical (i.e., tangible) and non-physical (i.e., intangible) environment*" [9].

The recognition for elements of ICH is consistently increasing [5]. Whereas before the 2003 Convention elements of ICH were supported and recorded, after the ratification many more elements were helped. Currently, the trend across global institutions of cultural

heritage has consisted in highlighting more the non-tangible aspects of heritage and the cultural heritage community. To the point that "*the distinction between physical [and intangible] heritage is now seen as artificial*" [12]. This is because the Convention scaffolds, with a legal instrument, what manners of intervention would be needed, and how they can be implemented. ICH is now understood as a vital link in the heritage chain that was not accounted for earlier.

Preserving intangible heritage, Bouchenaki explains, involves three aspects: Firstly, to support of practitioners and the community that created or carry on the tradition. Secondly, to put the associated tangible heritage in its wider context with the intangible elements; and thirdly, to translate intangible heritage into 'materiality' [5]. Sometimes this is a difficult task to accomplish with certain expressions of culture, such as large gatherings and festivals; but attempts can be made to 'materialise' them through video and interviews with organisers and participants.

As this paper shows, the development and the popularisation of digital technologies -that have become more affordable, portable, and count with user friendlier interfaces- has helped explore ways and opportunities of helping with these three aspects: supporting practitioners, contextualising tangible heritage within the intangible traditional practice, and "materialise" the intangible heritage to permit and facilitate its archival storage. This paper looks at digital motion capture technology to help preserve the tradition of Uillean pipe playing which was added to the ICH Representative list in 2017.

### 2.1. Inventorying

The preservation of intangible culture can be difficult. The UNESCO Convention has created three lists that help "raise awareness of intangible heritage and provide recognition to communities' traditions and know-how that reflect their cultural diversity": The Representative List (Article 16 of the Convention), the Urgent Safeguarding List (Article 17 of the Convention); and the list of Good Safeguarding Practices (Article 18 of the Convention) [14].

The first list -called the "Representative List" is a useful and necessary list of accepted elements that fulfil the criteria of Intangible Cultural Heritage. The purpose of listing them is to ensure respect for the practices by recognising them as well as creating a legal framework for assistance in the work of preservation as well as extolling them.

The second list, the Urgent Safeguarding List, refers to those aspects of the ICH in need of support to ensure the viability of the practice for this generation and others that follow. This list has been created to issue a warning about the imminent disappearance of an item of Cultural identity within a group. UNESCO [15] includes items in this list with the purpose of "*mobilising attention and international cooperation in order to safeguard intangible cultural heritage whose viability is at risk despite the best efforts of the community(ies) or the State(s) Party(ies)*" [16]. This means that UNESCO will take appropriate measures to safeguard the practice through resources and equipment, training, infrastructure, and expert knowledge.

The third list is dedicated to register Good Safeguarding Practises. "[it] *contains programs, projects and activities that best reflect the principles and the objectives of the Convention*" [16]. This list is for States, NGOs, Universities, and stakeholders to share good practice and success stories. The result of this list is a repository of knowledge about how to overcome challenges whilst preserving elements of ICH.

The UNESCO Committee needs archival evidence regarding the Element or Entity that is to be inscribed in any of the lists. This usually adopts the form of a video recording or a series of 10 photographs, and/or a (5–10 min) video with the community talking about the practice. The document comes together with a signed grant of rights to the video and photographs. The inscription process takes around 21 months from March Year 1 to December Year 2 [16].

According to the current regulation and guidelines regarding kinetic examples of ICH, the capture of the element with Motion Capture is not a requisite; but the effort to preserve and to register practices that could enter the list of intangible cultural heritage

has been appealing for the use of technologies and devices that help improve the quality of their recording and capture. There are some examples of technological use for the preservation of ICH; but the academic literature on the field is still scarce [17–19]. The special issue on *Mixed Reality and Gamification for Cultural Heritage* became a good point of reference for documented experiences [20]. The *i-Treasures* project is aimed at developing a repository and archive of digitally captured elements of Intangible Cultural Heritage. They reported that there were very few examples of Intangible heritage recorded through motion capture [21,22] and all of them were dedicated to recording dance. Other studies use Motion Capture to analyse dance elements [23–27] but few -if any- use the technology to capture musical instrument playing.

Aligned with this trend, examples of preservation of kinetic heritage such as the Virtual Dance Museum use large areas to invite dancers in, sometimes there are multiple (2–3) participants. Optical based motion capture is particularly good for this purpose: Optical systems use lycra suits which are often inexpensive to clothe multiple participants. The resultant dataset encompasses everyone together so that it is a continuous capture with participant interaction.

The experience introduced here is defined by the technological complexity of the devices involved (data-points must be combined on a 3D environment and their movement must be clearly recorded, this takes a high volume of digital data to be processed), the sensitivity and measurements required (the experience focused on capture finger movements playing at the rhythm of music), and by the expertise the process demands (the technology difficulties are combined by the scarcity of players available and the difficulty of playing the instrument. And this complexity goes both ways: playing the pipes is a dedicated and exceptional practice, and recording with Motion Capture suits, sensors, and rich animated digital data requires also a level of expertise.

These features make the effort for a collaborative work particularly relevant and ground-breaking, even if -as this is the case- this can only be thought of as an early stage of a broader unfolding process. The research team put some effort in trying to shift away from the technologies and focus instead on the "collaboratory" spirit [28]; but in these initial developments there was no particular purpose to the effort to co-develop pedagogical toolkits with the participants [28]; but only the background idea that collaboration is relevant for the consolidation of communities of practice [28].This article focuses on one of these technologies that should contribute to the preservation of the kinetic heritage by also incorporating the community of practice in the process. This is a collaborative design that involves the participation of the community of Uilleann pipe players and their views, combined with the exploration of the possibilities of motion capture (MoCap).

*2.2. Motion Capture*

There are several types of Motion Capture (MoCap) available. The first one is the camera-based systems such as marker based or marker-less type; and the second is the sensor-based systems such as Magnetic Motion Capture or Mechanical Motion Capture. Each one of these technologies has their advantages and disadvantages for the purposes described here. Another type of motion capture; is called volumetric motion Capture. It is like optical motion capture but captures the volume and clothes of the participant, this looks close to a 3D film of the event and can be further explored in relation to its applications in the Intangible Cultural Heritage context [29].

Optical Motion Capture is by far the most popular method of capture especially for film work. This method is based on the use of Infrared light reflected by spherical balls on the suit worn by the participant. The balks mark the participant's key joints or bones. These light-reflected points are tracked by cameras positioned around the periphery at various heights. The data is then brought together to show a 3D image of the participant in movement. If objects, or the actor, occlude the spheres, then other cameras within the 'rig' will see the sphere from another angle. However, occlusion is one problem for this set-up that needs to happen in a studio.

The rig is a condition for this optical system. It is a large volume room equipped with multiple cameras around (typically 20 or 30, but there can be many more). The cameras work together with the software to create a dataset that shows the participant's movements. Generally, this set up is very versatile and can capture multiple actors as well as objects (props-objects can have their own markers) and produce a relatively clean result which requires little post-production this is, after the recording the raw data resulting from the recording does not need too much clean-up.

An optical system can cost around $20–50,000+ depending on the complexity of the system required. It also demands a large room and an expensive camera rig as well as the software to amalgamate the multiple camera inputs. The set-up calibration can take about 5 min and involves measuring some vital measurements in the participant, such as Manus length, height, waist, and shoulder width. These measurements are then entered into the software. This technology can oversee small movements due to occlusion (for instance, the set up used on the experiment described in this article would not be particularly good for the capture of finger movements) [30].

Instead, sensor-based motion capture uses sensors on a suit or mechanism to assess where a participant is. This is the one used for the current experience: an Inertial Motion Capture suit manufactured by Rokoko Smartsuit v.1 with Smartgloves v1. These suit and gloves use IMU (Inertial Measurement Unit) [31] sensors to calibrate the human form, and to capture movement and pose. This suit uses multiple 19–6 axis IMU (Inertial Measurement Unit) sensors [32] embedded within the fabric, and the data they collect is then brought together by the accompanying sensor fusion software [33]. In this case, the additional gloves which also use IMU sensors were added to capture the finger movements of the participant.

Inertial Motion Capture is lighter and portable than the Optic. Captures can take place anywhere (as long as there is a power outlet for WiFi as well as low magnetic interference for the inertial sensors). Accuracy is moderately good and the MoCap data requires little clean-up aside from some 'sliding' of the actor. In comparison, this setup is relatively inexpensive with that of optical motion capture at $2–3000. A community can thus purchase and record their own motion capture sessions rather than having to rent a full studio. This also means that recordings can be done directly in the field and the recording is available to view immediately afterwards. This is a feature that also adds to the preservation purpose because it is less invasive than the optical capture that needs the performance to be moved away from its context of practice onto a specifically enabled studio-space.

Inertial Motion Capture is a technologically sophisticated piece of hardware and software equipment that solves several problems in the budget end of Motion Capture. The commercial main use for this equipment responds to the need for a democratisation of Motion Capture in applications such as budget film making, and games. Therefore, this equipment and its uses can easily be re-oriented to encompass capturing elements of Intangible Cultural Heritage for Communities who need a low-cost solution to create their own archives and inventories of their own practice.

### 2.3. Uilleann Pipes

The Uilleann (Uillean in Irish means 'Elbow') pipes can be traced back to 1743 with the publication of John Geoghegan's book "*Compleat tutor for the pastoral or new Bagpipe*" [34] in London. This pipe was partially recognisable as the Pastoral pipe [22] a close relative to the Northumbrian small pipe [35]; but it continued to evolve over the eighteenth century, to become what we know today as the Uilleann pipe: the pipe powered by an elbow bellows. Culturally the Uilleann pipe was seen as a shared tradition between Britain and Ireland [21] due to the evolution of several distinguished types of pipes, Uilleann, Union, Pastoral, Northumbrian, and Scottish Lowland.

The instrument includes a regulator (a closed pipe with a set of tone-holes), drones with regulators (which provide self-accompaniment) and a chanter [1] (see Figure 1). In the National Museum of Ireland there is a set which is said to have belonged to Lord Edward

Fitzgerald with these features hall-marked to 1768 [1]. It was about this time that players decided to sit to play these pipes.

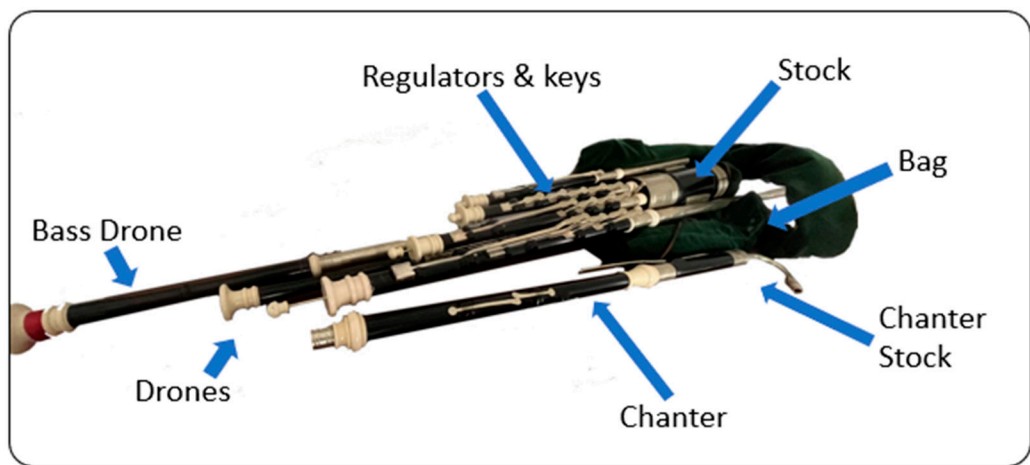

**Figure 1.** The Uilleann pipes (Full concert set) used by the participant in the Motion Capture session.

Uilleann pipes have cultural significance in Ireland, and their performance has almost died out several times [36]. More recently, however, there has been a strong cultural effort to keep the practice of playing the pipes from dwindling [37] especially empowering of the community as formulated in the UNESCO 2003 Convention on Intangible Cultural Heritage. This involves supporting key stakeholders and gatekeepers within which the element or practice resides. In this case, Na Piobairi Uilleann (The Uilleann pipers society) represents pipers and pipe makers across the world (not just in Ireland). The Society was formed in 1968 by Uilleann pipe players and makers [34–36] Paddy Maloney, Leo Rowsome, Seamus Ennis, Breandan Breathnach. Since then, the Pipers Society has taken very active steps in encouraging tutelage, scholarships, practice, archiving, and dissemination of Uilleann pipe playing, so much so that the playing of Uilleann pipes is no longer in danger of disappearance as it once was; and it enjoys international and heritage related recognition. The Playing and Manufacture of the Uilleann pipes were added to the UNESCO's Representation List of Intangible Culture Heritage in 2017 [38].

## 3. Materials and Methods

This paper reports the experience used as a pilot for a longer-term study into how the piping methods can be captured and how the community can gain use of this data to archive, and to improve the playing techniques and their teaching within the community. This paper is part of a broader effort to connect the possibilities of Inertial Motion Capture as a tool for the development of collaborative design practices that help the capture, contribution, and preservation of ICH practices.

At this early stage, the experience was operationalised to provide outcomes along three distinct aspects: first, the Motion Capture technique in itself; then the impression of the experience on the players; and last the results of the practice by incorporating discussions about future possibilities and training options. These three strands also defined the procedure.

The first contact was made through Social Media. A message was shared via the Irish traditional music scene, and the NPU (Na Piobairi Uilleann) Association who put out a call for pipers to help in the research. The text mentioned that researchers were looking for a Uilleann piper to be Motion Captured (MoCap) for this project. This was a pilot study initially designed to feed the dataset of a PhD. Previous knowledge about the nature of piping and its practicalities in this case was fairly sparce to begin with. Also, this is the first time that this motion capture equipment has been used to research the performing of

players within the piping community. So, the lack of knowledge was here two-way: early research scholarship, and no experience among the performer.

A professional piper (originally from County Tyrone, Ireland; but currently living in the UK) answered, and they were invited to participate in a meeting set up in Sheffield. The recording of the player motions took one day to be completed.

The process of data collection was therefore organised in three stages. First the accommodation of the player and the introduction to the process. The piper worked with the researcher to facilitate a better set of data: this involved discussing the use and fitting of the suit, understanding of the space, and of the recording processes of the motion capture. The agreement also included that the piper would play five jigs or reels, stopping after each one. The airs played would vary from some slower to faster tunes; this would help challenge and assess the limitations and possibilities of the technological equipment. This process included the sharing of real-time playback of the motion capture recordings.

The collaborative design also led researcher and piper to agree that between tunes the sensors would be re-calibrated, and the pipe re-tuned. During the stop time the batteries of the equipment were also checked to ensure good levels, and the software was monitored to ensure a good output. This should help avoid unexpected software artefacts -bad synchronisation outputs- in the results, and a comfortable good sound for the piper.

The second stage of the data collection process was the more technical part. It included setting up the equipment and calibration. The equipment was set up in the studio at Hallam. Gloves and Suit needed to be recognised by the smart-suit software (this is called the 'handshake'). Only after that initial synchronisation the process could be recorded in the 'Smart-suit' software. The event was also video recorded with a camera was locked off in front of the participant. The video recording was to be used later to synchronise with the footage of the Motion Capture recording.

Hips, shoulders, legs, and torso of the participant were also measured to develop later the creation of a reasonable software avatar. The computer generated then an Actor of the approximate size of the pipe player. This projected Actor dictates the length of the joints and the distance from each other. The Actor was then joined together with the gloves to create the complete software produced avatar (see Figure 3).

This process of calibration of the participant in the system needed to happen before the recording. This meant that the participant had to stand with arms by the side, feet facing forward, and not moving for 3 s. After that, the pipes had to be tuned again.

The actual recording could begin after that.

The third stage of data collection is the interview. This is more an open conversation about the whole process where player and researchers comment, record, and share about the comfort of playing with the suit, or the experience, and the possibilities of the technology as training tool and as archivistic preservation. The interview is also a way to ensure that participant and researcher could review and comment on the results of the MoCap session: including the revision of the MoCap footage to identify further steps and improvements. As part of this conversation the participant was also invited to review the footage of another piper taken earlier in the year in County Waterford, Ireland.

The analysis of the data also required stages: on one side, the interview is considered also as part of the analysis, it provides helpful insight of the experience and resonates with the principles of collaborative design; and on the other side part of the analysis consists on the whole technical process of cleaning and arranging the MoCap data.

After the session was completed, the footage from the Motion Capture session, as well as video footage were exported to be cleaned (as required) and further analysed. (Figures 2 and 3). The data clean-up process is a necessary process. The motion capture footage often has errors. In the case of Marker based Motion Capture these errors tend to be more numerous, because the points on the suit can be occluded from the cameras. In the case of Inertial MoCap these errors tend to be *sliding* or generate *drift* errors of the whole body. They occur because the capturing system works in the absence of fixed points on the suit that anchor outside itself to a specific place in the physical world.

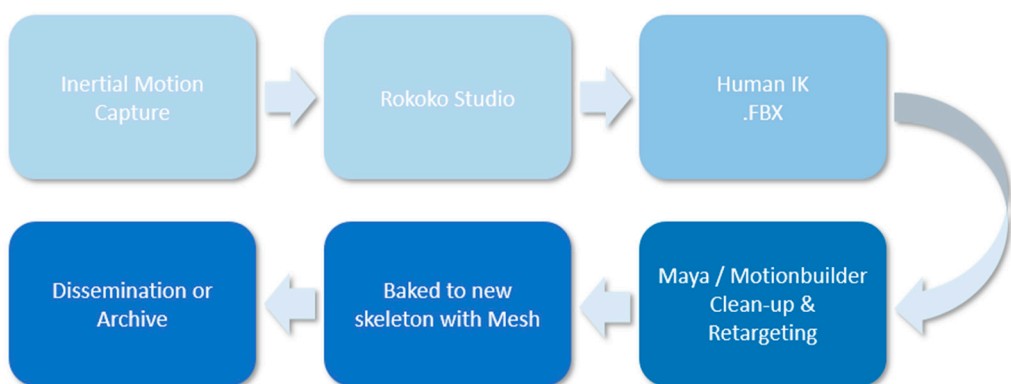

**Figure 2.** This is the flow diagram for the process of capturing the participant motion.

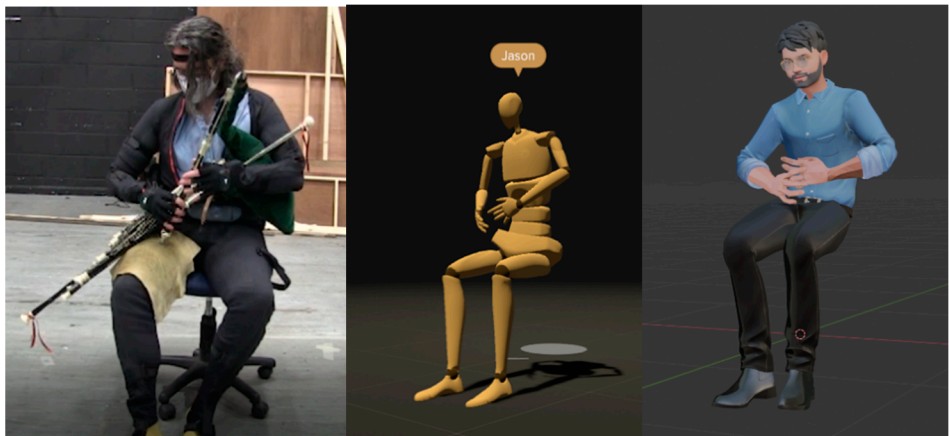

**Figure 3.** Left the participant in the Mocap suit, centre the live recording in the editor, right the rendered version in Blender (please note these are not at the same stage in the tune).

To check and clean it, the data needs to be exported from Rokoko software as an .fbx format. This is filmbox, a native motion capture file system. That data is later re-imported into Autodesk Motionbuilder so that the drift errors can be properly eliminated. There are less drift errors if after every jig or reel the machine has been re-calibrated. The data is then imported in the form of a point cloud which can be cleaned up prior to being attached to a skeletal rig. The Mocap footage is useful as is in the Rokoko editor but to increase its possibilities of use, it needs to be exported to Motionbuilder or another 3D package - this part takes time and skills because it needs to be attached (skinned) to a skeleton. The skeletal rig can then be transferred to a mesh (this process is called 'skinning'). This mesh can then be output or disseminated.

These three strands of data collection and of data analysis helped provide a round view of the possibilities of inertia motion capture for the piper community. The next section details some of the findings.

## 4. Results

This section is arranged along two lines. First, the findings and outputs associated strictly with the data and the technical process of MoCap and its features; second, about the feedback and inputs and conversation with the player around the experience of the process; and finally, Section 4 is a discussion about the aspects related to the collaborative design, its implementation, and limitations. These aspects will then inform the conclusions.

### 4.1. Findings Regarding the MoCap Data and Features

The overall results were very good. Figure 3 shows the translation of the MoCap process from the actual player to the Blender-rendered avatar. Note, from instance, that

the suit needed to remain open in the chest area as it did not fit the player size; or the particularly enhanced hands in the avatar, that are also relevant for the precision of the fingers during the performance. A small extract of the practice can be seen online [39].

There were anomalies that had arisen in the data which changed the motion of the captured character. The process helped identify three critical moments: the first one is the synchronisation between the movement data collection and the sound; the second critical moment is the need of editing of data that was not properly captured as part of the synchronisation issue; and the third is the lack of consensus between cameras that requires systematic recalibration.

The synchronisation of the data with the sound file can be a problem because the sound file is recorded separately to the MoCap file. This does not happen with video because its native soundtrack is recorded at the same time as the images. This means that the soundtrack needs to be further worked and synched up at various cue's (for example the hand clapping gesture between each take) but that can also lead to problems, such as frame rate discrepancies that might lead to a progressive separation between sound and motion.

Most of the generated data can be 'cleaned up' to some extent, the cleaning process might affect the authenticity of the results; but this is generally minor, and it is a consistent issue across all motion Capture systems. The way around these issues might require editing out sections (taking out the tuning of the instrument, which is an interesting phenomenon and worthy of study) or moving hands or feet to where they need to be. These editing operations do not change dramatically the nature of the movement and the video reference is vital (specially to keep track and synchronicity with the sound recording).

This is a usual issue embedded in the same nature of Inertial Motion Capture: each sensor is responsible for its own location against the others in the chain; they do not have a "consensus" of cameras as it happens in the case of Optical systems. To solve this, the best option is to recalibrate after each 'take'.

Altogether, and on viewing the results there are two issues that should be addressed: accuracy and framerate. Both issues are technical and beyond the extent of this study; but the first one, involves the suit itself, which as the technology is in its infancy. It will likely improve with further iterations of the technology and the increasing the accuracy of the IMU sensors. One of the reasons for this is that the suit used on this occasion is toward the budget end of the available suits. This is a trade-off as one of the criteria for this study would be to use a suit that a community might be able to afford for themselves. The second issue involves capturing at a higher framerate from the equipment itself again this can be a suit or Wi-Fi issue. In either case a more expensive set of equipment may give better results but would be less likely to be purchased by a community wishing to record elements of ICH for themselves.

### 4.2. Conversation with the Player

From the perspective of the performer, the suit and the wearability also showed some issues as it was recorded in the interview after the collaborative experience: "*The stretchiness of the fabric is great*", and the overall comfort and utility were praised: the Smart gloves, [which are fingerless] "*I was very glad they were fingerless because I wouldn't have been able to do much on the chanter otherwise really*", this has particular relevance to feel the keys of the regulator: "*the gloves were great but not being able to feel where the keys are, made it quite tricky to get the regulators because they are tiny keys you've got to be fairly specific about how you play them, you know*". In relation to the suit and how it feels, the player also mentioned "*I felt like I was being pulled back a little bit*". Also, because the straps of the suit "*occasionally would get in the way and were trapped by the Regulator keys*". The size of the suit matters and in this case, the performer's chest did not fit well; the suit had to be worn open (see Figure 3). The suit sizing is a problem, and a solution could be better achieved with a more flexible fabric. This however did not affect the motion capture as the participant did not have to move around the stage; but it would have if the recording involved choreography, for instance.

This will have to be a consideration in future experiences: ensure that the size of the suit fits the participants before the recording session. The recording could be instantly viewed after the session, and the performer expressed its interest and value: "*Then you've got your video on top of that . . . , but this is like another level of data which is really exciting*".

Upon watching the recording the speed of the capture, especially with faster tunes also raised some comments "It would work for a slower tune like a jig or an air or something, but when you are playing reels or whatever you want to see more [framerate]". The piper, in this sense, noted that the tempo of the capture was off in some circumstances.

This high level of detail of the animated data immediately helped the performer review his own practice and style: "*Instantly I thought: Oh I'm doing this wrong or that wrong. Not something I would have noticed, sometimes I used to sit in front of a mirror and play to check my posture and everything or seen a video of it, . . . you can't really tell. That was an eye opener alright yea*". This level of detail offers a good opportunity for further learning and further reproducing the culture. It is an opportunity for learning and for training. Students can check their own style when they play. And it is a good way for performers to assess and improve (but also to share) their own work.

And with the training and the detail, this technique brings some advantages and opportunities in terms of archival recording. When the participant was asked to speculate about the archival utility of the model they said: "*A lot of it, you miss from those kind of recordings or even with a video, or seeing someone in real life. There is a performative aspect as well which is what you really get, even though it's a computer model*".

The detail of the data is illuminating beyond the specifics of the performance. The image has a density that highlights the idiosyncrasy of the player and performer with enough detail for an expert eye to recognise their own "accent":

The performer identified himself as a player even if the image was still in blocks: "*When I was looking at it ( . . . ), it was definitely me. Those are my piping mannerisms*". And then he explained: "*a lot of pipers are losing that localised accent, like piping accent in their playing, years ago you could tell where somebody was from by how they played, like he's an Armagh player or this is the Wexford style or a Clare style and now everyone's got access to everything and all the time so people pick and choose and we end up with this weird homogenised idea, like a big stew of different styles, instead of a localised style, I'm quite proud of my piping accent*".

The participant was very impressed with the footage of his fellow piper (the participant from Waterford) and detected his 'accent', that is movement and playing style which is influenced by the tutor/pupil relationship:

> "*Absolutely! his stance and how he plays he's got a good bounce to him! And you can see that in the [Waterford pipers], they have that rhythm bounce to their playing 100% capturing snapshots of regions with their movements.*
>
> *That's great! he's got a good bounce to him hasn't he?*
>
> *Can you tell from that, that he's a Waterford piper? (Interviewer)*
>
> *Yes, Certainly! He's very animated. Really captures him*".

Indeed, the regional "accents" of the pipers are a feature usually gained through tradition and personalised teaching-pupil transmission. The growing presence of social media resources -such as YouTube- from where many new pipers are learning the basic skills beyond Ireland across the globe, progressively erodes these traits or accents in the playing of the instrument and homogenizes the whole teaching and practice. As much as the rising popularity of the instrument can be seen as a good thing to ensure the continuity of the tradition, this homogenisation process might end up with the loss of the piper's accents.

These comments from the performer right after the review of his own performance are very valuable inputs because they expand the discussion beyond the specific practice of playing the pipes towards the richness and diversity of the context within which this instrument is played: the accents are marks of diversity, as much as they are marks of identity and -for those who can read them- they are also marks of geographic difference.

All these are fundamental opportunities to discuss the third aspect of this research: the motion capture technique as a tool that helps connect the practice with the broader context of the cultural environment, and the possibilities of expanding this area of research on collaborative design.

## 5. Conclusions: Combining Collaborative Design and Inertia Motion Capture

In this paper we described an experience of collaborative design between Inertial Motion Capture and the recording of the work of Intangible Cultural Heritage listed Uilleann pipe playing. The specialism required to play the Uilleann pipe, and the specialized skills and equipment necessary to use the technique of Motion Capture made of this combination a particularly interesting occasion, and a relevant opportunity. The conclusions obtained from this experience can be organised along two strands: first, the review of the features of Inertia Motion Capture for the recording of intangible culture heritage practices such as instrument playing; and second, the report of how this collaborative design experience between Inertia Motion Capture and the Uillean pipes tradition has provided the ground work for the development of a more specialized research program.

### 5.1. The Features of Inertia Motion Capture for the Recording of Intangible Heritage Practices

Inertia Motion Capture offers three clear advantages that align with the principles of the recording of intangible heritage practices such as instrument playing: the devices and set for Inertia Motion Capture are portable, the equipment is inexpensive, and the recording provides a high amount of detail at 3D scale. These three advantages are very significant if applied to ICH because they connect well with the three requirements established in relation to the preservation of Intangible Culture Heritage: the support to the community of practitioners involved in the tradition; ensuring continuity between objects and practices and the wider context of intangible elements; and enabling possibilities of materially translating the heritage to make it preservable [5]. Inertia Motion Capture, thus is a tool sensible and adaptable to the context -location and value- of the cultural practice and an opportunity for Intangible Cultural Heritage.

Indeed, the portable Inertial Motion Capture equipment connects with the community. It is true that it still needs some space and weights; and that it involves suits, sensors, batteries, and recording devices such as a laptop, a video camera, and the complementary sound equipment; but the full pack can be carried in a few handbags, and it can be brought and installed in the locations of practice. In contrast with other techniques with similar results that require having to ask the practitioners to move to specifically built recording studios (as it happens with optic motion capture technique). Also, the suit itself is relatively comfortable -admittedly, the sizing is a problem and could be better achieved with a more flexible fabric or a full body mesh. This portability of the technology kit allows for the closeness of the recording to the community of practice and to the locations of the tradition; which also facilitates the participation and the collaborative design involving the community in the process of organisation, planification, production, recording, and analysis of the quality of the results.

Also, the equipment for Inertia Motion Capture is inexpensive. The equipment and set up costs are relatively low in front of other similar techniques. It is not unthinkable for communities of practice to acquire or rent the equipment and the technical support to record elements themselves in-situ. In comparison, the set-up costs and maintenance for an Optical Motion capture system are considerably higher both in terms of acquisition and of maintenance. Regarding the skillset necessary to perform the capture, Inertial Motion Capture demands some technical knowledge to clean, fully render, and to disseminate the data; but also, the specific human skills can also be delegated: the skills for recording can be acquired by the community members and then the skills for rendering and cleaning can be obtained from elsewhere, substantially reducing the costs of the whole experience. And furthermore, the whole process can be done with open-source software, which again benefits the idea of a community driven archivism and aligns with the UNESCO vision and

principles. This inexpensive technology -that can be rented or acquired with little skills and open source- supports the possibilities of the communities to own their own production, to ensure that there is a close connection between the practice and the recording, the practice, the environment, and the objects.

The third advantage of Inertia Motion Capture, in front of other techniques -such as video, or optic motion capture-, is that Inertia Motion Capture deals better with smaller detail. The technology is more suitable to larger kinetic motions, and at small scales can lose accuracy, but the experience showed that the gloves allowed the capture of outstanding detail data of the participant's finger motions. The sole capacity of capturing the finger movements of the player -thanks to the glove and its sensors- makes the animation precise enough to capture the local accents and the idiosyncratic styles of the players. This is a level of detail almost unperceivable at naked eye for anyone not expert in the field; and it is very difficult to capture with any other technique. This technology, then, helps enhancing those particularities and opens a field of possibilities for further study, implementation, and outreach. Such capacity of capturing detail -to the specific movement of the fingers- translates into storable data the recording of a movement that would otherwise not be perceived by most of the public, and that would be difficult to capture at this level of precision by videos or by any other movement capture systems. Such detail capture is a form of materialisation -into recordings- of the collaborative experience, ensures the archival possibility of this material which is the third of the requirements for Intangible Culture Heritage.

These three achievements portability and community engagement; inexpensive technology and connection between the objects, the practice and the symbolic context; and the material possibility of preservation of the experience to the very specific detail make inertia motion capture a good candidate as a technology to collaborate and to operate within the preservation and archival tasks of intangible cultural heritage.

This type of recording opens accidentally another relevant pending discussion (that cannot be tackled in this occasion): the opportunity of sharing and spreading the video and animated results out and about through multiplying social media platforms thanks a detailed 3D video is a chance for diffusion and for social outreach; but it also raises the issue of the legal ownership and rights of the piece; and with this, the circuit of the authorial intellectual property context. Classically, copyright of folklore was one of the earliest protections of international law concerning elements of Intangible Heritage and it served as inspiration for UNESCO's 'Living Human Treasures' system [29]. Overall, in his chapter on cultural heritage Shyllon argues that there is "inadequate legal protection of cultural heritage" [40] as digital simulacra can easily be made and disseminated, even though the conventions are essentially a legal framework. This aspect should inspire, rather than deter further research in the field; and explore how this can be solved and how the participant can be granted legal rights over the Motion Capture of their performance; but even if this discussion cannot be held right away here, it is fundamental that this question it is raised on this occasion.

### 5.2. Collaborative Design and Inertia Motion Capture for the Uillean Pipes Tradition

The opportunity of designing this experience as a collaborative practice that incorporated the views, suggestions, and contributions of a piper as participant/performer all along the process was vital for the quality of the outputs. The collaborative spirit helped immediately to define and steer the recording process and to anticipate and solve technical problems emerging before and during the performance. This collaborative spirit also helped with sound analytical output: the presence of the piper enhanced the ongoing process of analysis after the rendering by providing expert observations on the style, speed, and quality of the process. The very specialised comment and reading of the images animated from the motion capture would have never appeared without the performer's active collaborative engagement in the design process and in the analytical reviewing of the data collected. This process happened in real time and informed the whole collection process.

The trained gaze of the player helped identify features in relation to what he called "the accent" allegedly a regional mark that is imprinted in the playing style. The aspect of the pipers' accents confirms the efficiency and robust possibilities of Motion Capture, but it also insists on the need of collaborative designs to let this analysis emerge. Furthermore, the notion of the regional accent allows, for a rich anthropological exploration that opens now as a range of opportunities that will guide the further development, improvement, and implementation of further specific research. The collaborative design approach is now growing into a research program that will be specifically designed to identify and to map these features; and to contribute to an already existing informal debate among the players in the field.

This is a prominent outcome that epitomizes the possibilities of the collaboration between performers and researchers; and it is a strong indicator that collaborative design is the way to go to improve the research results in this field. Indeed, Inertia Motion Capture is thus effective at capturing some elements of Intangible Cultural Heritage, and a relatively quick instrument to disseminate them and to archive them. And still, used within the frame of collaborative co-design this recording technique can help support a rich range of community linked future possibilities: such as improving the understanding of the community of practice, and as a powerful tool for educational purposes [41].

**Author Contributions:** Conceptualization, P.I.M. and J.R.R.-A.; methodology, P.I.M.; software, P.I.M.; validation, P.I.M. and J.R.R.-A.; formal analysis, P.I.M. and piper; investigation, P.I.M.; resources P.I.M.; data curation, P.I.M.; writing—original draft preparation, P.I.M. and J.R.R.-A.; writing—review and editing, J.R.R.-A.; visualization, P.I.M.; supervision, J.R.R.-A. All authors have read and agreed to the published version of the manuscript.

**Funding:** This research received no external funding, only internal support from the Media Arts and Communication Department at Sheffield Hallam University.

**Institutional Review Board Statement:** This study was conducted according to the Sheffield Hallam University and approved by the Ethics Committee ID: ER38258538; and the broader project was approved by the ethics committee at the University of Leicester, United Kingdom.

**Informed Consent Statement:** Informed consent was obtained from all subjects involved in the study.

**Conflicts of Interest:** The authors declare no conflict of interest.

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
