# Peer review of "Collaborative Design in Kinetic Performance: Safeguarding the Uilleann Pipes through Inertial Motion Capture"

_mti, doi:10.3390/mti6110097_

Round 1

Reviewer 1 Report

The article is prepared in a concise language, understandable to the reader. Many aspects have been explained. It would be worth introducing the reader to the topic (elaborate on a few issues, as these may be unclear for people who are unfamiliar with this topic). The numbering of the used literature in the text should be corrected! It would be worth adding useful conclusions.

the results obtained should be shortened and summarized. Many of them are redundant. Applications should be kept to a minimum. In terms of volume, they do not match the whole. too many requests by volume!

Author Response

Thank you very much for the feedback. It has helped us to go through the article again and to review the text to ensure it is easier to understand and clearer overall. The literature numbering was also corrected and all aspects mentioned clarified. However the practice driven experience described on this occasion leads towards the detailed description of results to help further practitioners and to establish a starting ground for further deeper and probably more substantial research outputs. At this stage, this is a very successful start.  

Reviewer 2 Report

The paper describes a design protocol where the authors explored the chance to use motion capture technologies and 3D Blender software for visual feedback to study Uilleann pipes. The article was well written, presenting ad adeguate state of art. The methodology is well explained and the topic results interesting for the reviewer. Nevertheless, the sample presented is really small, I suggest to extend the study involving other musicians in the recordings. I would also suggest that the authors better explain the processing of raw motion capture data, to be able to create the final rendering in Blender 3D, for the benefit of those who are not experts in mocap systems. Finally, I would suggest that the authors create a few seconds of video that they can show to further emphasize the effectiveness of the methodology employed.

Author Response

Thank you very much for the review and comments. 

We have now been through the paper to make sure that the process is clearer and to ensure that it is still readable and explanatory. We have improved the reference list and checked spelling and grammar. 

Also following the suggestion of the reviewer we have also added a video of the experience on Youtube and the link in the bibliography:  https://www.youtube.com/watch?v=wWUQKr2Ld0o

We appreciate that there is a lot to improve in the whole process of data collection, and probably in the future the comparative exploration will be done through practice and demonstration, but at this stage this could only be done based on the knowledge of the medium and expertise of the authors.